# Emerging Roles of Cholinergic Receptors in Schwann Cell Development and Plasticity

**DOI:** 10.3390/biomedicines11010041

**Published:** 2022-12-24

**Authors:** Roberta Piovesana, Adam J. Reid, Ada Maria Tata

**Affiliations:** 1Départment de Neurosciences, Université de Montréal, Montréal, QC H3T 1J4, Canada; 2Groupe de Recherche sur le Systeme Nerveux Central, Université de Montréal, Montréal, QC H3T 1J4, Canada; 3Blond McIndoe Laboratories, Division of Cell Matrix Biology and Regenerative Medicine, School of Biological Sciences, Faculty of Biology, Medicine and Health, The University of Manchester, Manchester Academic Health Science Centre, Manchester M13 9PT, UK; 4Department of Plastic Surgery & Burns, Wythenshawe Hospital, Manchester University NHS Foundation Trust, Manchester Academic Health Science Centre, Manchester M23 9LT, UK; 5Department of Biology and Biotechnologies “Charles Darwin”, “Sapienza” University of Rome, 00185 Rome, Italy; 6Research Centre of Neurobiology “Daniel Bovet”, “Sapienza” University of Rome, 00185 Rome, Italy

**Keywords:** neuron–glia interaction, neurotransmitter receptors, acetylcholine, muscarinic receptors, nicotinic receptors, differentiation, myelination

## Abstract

The cross talk between neurons and glial cells during development, adulthood, and disease, has been extensively documented. Among the molecules mediating these interactions, neurotransmitters play a relevant role both in myelinating and non-myelinating glial cells, thus resulting as additional candidates regulating the development and physiology of the glial cells. In this review, we summarise the contribution of the main neurotransmitter receptors in the regulation of the morphogenetic events of glial cells, with particular attention paid to the role of acetylcholine receptors in Schwann cell physiology. In particular, the M2 muscarinic receptor influences Schwann cell phenotype and the α7 nicotinic receptor is emerging as influential in the modulation of peripheral nerve regeneration and inflammation. This new evidence significantly improves our knowledge of Schwann cell development and function and may contribute to identifying interesting new targets to support the activity of these cells in pathological conditions.

## 1. Introduction

During the development of the peripheral nervous system (PNS), neurons and glial cells form an integrated system, which is provided by an exclusive and reciprocal molecular network interaction and is able to maintain an efficient homeostasis, regulating the development and the physiology of both components [1,2]. During neurogenesis, glial cells provide distinct paracrine signalling to support neuronal survival, define the molecular domains of the axolemma and axonal diameter, and maintain an appropriate concentration of neurotransmitters in the neuronal environment [3,4]. Axons, by contrast, express and release molecules regulating glial cell proliferation, survival, differentiation, and myelin formation. In adulthood, neuron–glia interactions are also required for the preservation of axonal function and myelin integrity [5,6]. The alteration of neuron–glia communication causes severe dysfunctions that could compromise the PNS physiology. Schwann cells (SCs), the main glial cells of the PNS, are involved in axon myelination and in the maintenance of PNS homeostasis [7]. Several molecules and mechanisms involved in SC development and function have been identified [8,9] and, among them, the role of neurotransmitters in the communication between axons and glial cells has largely emerged [10,11,12]. In this review, we summarise the data reporting the functional role of neurotransmitters in the regulation of glial cell development and functions, highlighting the role of ACh and its receptors in the PNS. In particular, we describe the role of M2 muscarinic cholinergic receptors in SC differentiation and the effects mediated by α7 nicotinic receptors on SCs in peripheral nerve regeneration.

## 2. Neurotransmitters in Neuron–Glial Interaction

Neurotransmitters are defined as neuromediators released at the synaptic level which regulate cell communication between neurons and their specific cell targets. The expression of neurotransmitter receptors on the membrane of neural precursors during the early stages of neurogenesis and prior to synaptogenesis suggests their possible contribution to the ontogenesis of central (CNS) and peripheral (PNS) nervous systems [13,14]. In vitro and in vivo studies demonstrate that neurotransmitter receptors are expressed by glial cells and are involved in morphogenetic events during glial cell development and, additionally, contribute to the control of glial cell function in physiological and pathological events [7,11,15,16] overall when released in extra-synaptic regions [17,18,19]. 

A summary of neurotransmitter roles in glial cells is reported below and in Table 1.

***Purine receptors***. Purine receptors are divided into adenosine (P1) and adenosine triphosphate (ATP) (P2) receptors. Purinergic receptor subtypes are widely expressed in the CNS, contributing to the synaptic transmission, neuromodulation, and neuroinflammatory response [48,49,50]. Interestingly, Stevens and Fields proposed contradictory roles of adenosine in glial cells through the promotion of glial cell differentiation and myelination in the CNS in contrast to the arrest or delay of these functions in the PNS [7,51].

Neuron–glial signalling is mediated by ATP in PNS through metabotropic P2Y and ionotropic P2X receptors expressed on SCs. In the CNS, adenosine enhances oligodendrocytes differentiation from progenitor stage and later promoting myelination through adenosine (P1) receptor activation [20]. These observations suggest that purine-mediated effects are dependent on receptor patterns expressed in the glial cells and on their intracellular regulatory cascade.

In the PNS, P2X4R is generally localised in the lysosomes of SCs, but its expression on SC plasma membrane surface is mediated by tumor necrosis factor-α (TNF-α). In vivo experiments showed that P2X4R expression in SCs is strongly upregulated following nerve crush injury, where it accelerates motor and sensory functional recovery and nerve remyelination via BDNF release [25]. Furthermore, P2X7 receptor (P2RX7) knockout mice showed an increase in mRNA levels of myelin proteins (i.e., myelin protein zero (P0) and myelin-associated glycoprotein (MAG)), demonstrating that purinergic receptors caused a reduced expression of myelin protein, promoting the formation of Remak bundles in the sciatic nerves [21]. 

***Glutamate receptors***. During neurogenesis, elevated concentrations of the excitatory neurotransmitter glutamate are present in developing areas of the brain. Besides its conventional role in neurotransmission, glutamate regulates neuronal circuitry and cytoarchitecture through the regulation of neuronal differentiation [52]. Glutamate receptors are classified in metabotropic G-coupled receptors (mGluRs) and heteromeric ligand-gated ion channels, iGluRs [16]; both are also expressed by glial cells [52]. 

Rat and mouse sciatic nerve studies have shown that dehydrogenase glutamic acid decarboxylase (GAD67) and glutamine synthetase are expressed in in vitro SCs and in nerve slices [34,53,54]. In vitro studies have also shown that, through the expression of excitatory acid transporter-1, SCs take up glutamate from the extracellular environment, and after metabolization they synthesise new neuromediators for release as autocrine and paracrine signalling on neighbouring SCs [27,55].

mGluR2 activation promotes an immature phenotype via NRG-1-induced ERK phosphorylation, enhancing cell proliferation; on the other hand, its downregulation upregulates EGR-2/Krox-20 and myelin basic protein (MBP) together with the decrease of c-jun expression [32].

***γ-aminobutyric acid**(GABA) receptors***. Following its release from nerve endings, GABA acts via ionotropic GABA-A and GABA-C receptors or the metabotropic GABA-B receptor. In CNS, the trophic function of GABA receptors seems to be related to the biology of myelin [33]. However, the action of GABA is not only limited to the CNS, but it is also active in peripheral tissue. Several subunits of the GABA-A receptor (i.e., α2, α3, and β1–3), as well as GABA-B1 and GABA-B2 receptors, are expressed in rat sciatic nerves and cultured SCs [56,57]. SCs not only express GABA-A receptors, but they are able to synthesise GABA, as suggested by the expression of the cytoplasmic GAD67, whose levels are under the positive control of GABA-A receptor activation [34]. Interestingly the knockout mice for GABA-B receptors showed myelin alterations, associated with changes in myelin protein expression (e.g., P0 and PMP22) in the sciatic nerve, and an increased number of small caliber myelinated fibers and an enrichment of the subpopulation of small DRG neurons [35,57].

## 3. Acetylcholine and Cholinergic Receptors in Glial Cells

Acetylcholine was the first molecule identified as a neurotransmitter; its detection in unicellular organisms (i.e., bacteria, protozoa) and primitive plants suggested that ACh may be one of the oldest signalling molecules, highlighting that ACh was widely distributed in primitive organisms before its detection in the nervous system [17,58,59], where it acts as a regulator of nervous system development [60,61]. The cholinergic system is also expressed in several non-neuronal tissues where ACh is not necessarily derived from cholinergic innervation, but it can be synthesised by different cell types (e.g., lymphocytes, keratinocytes, stem cells, and lung epithelial cells) and, upon binding cholinergic receptors, may activate autocrine and/or paracrine signals [17,62,63,64,65], thereby modulating cell growth, survival, and differentiation [66].

ACh is synthesised by the enzyme choline acetyltransferase (ChAT) using choline and acetyl coenzyme A as substrates; then it is released in synaptic and extrasynaptic regions [19] and rapidly degraded by acetylcholinesterase (AChE) into choline and acetic acid [44].

ACh receptors are grouped into two large families according to their structures. The name of cholinergic receptors depends on their natural agonists: muscarinic metabotropic receptors and nicotinic ionotropic receptors. In CNS and PNS, both receptor types are present on neurons and glial cells [38,40,67,68] and mediate multiple neuronal/glial physiological events, such as neuronal plasticity and synapse function associated with learning, cognition and memory, and glial differentiation and myelination. Indeed, nicotinic and muscarinic receptor expression is significantly altered in various CNS pathologies, such as schizophrenia and Alzheimer’s and Parkinson’s disease [69,70].

## 4. Muscarinic Receptors

Five different subtypes of muscarinic receptors, encoded by different genes, were discovered in mammalian and non-mammalian species. Muscarinic receptors are activated by the alkaloid muscarine from the mushroom *Amanita muscaria* and blocked by ‘*Belladonna*’ alkaloids (e.g., atropine and scopolamine). They belong to the 7-transmembrane G-protein coupled receptors family (GPCRs) [71]. Their structure shows a high degree of homology with more variability limited to the carboxy and amino terminals and to the third cytoplasmic loop [72]. This homology of sequence has limited development of pharmacological treatments and new selective ligands to target specific muscarinic receptor functions [73].

Muscarinic receptors are classified according to their G protein coupling:M1, M3, and M5 receptors, coupled with G_q/11_ protein, are able to stimulate inositol trisphosphate (IP3) hydrolysis and intracellular calcium mobilisation. They can also regulate phospholipase A2 and phospholipase D.M2 and M4 receptors are coupled to G_αi_ protein which inhibits adenylate cyclase activity, decreasing the intracellular levels of cyclic AMP (cAMP) [74]. Moreover, they can modulate the activity of K^+^ channels.

Additionally, muscarinic receptor subtypes can stimulate small G proteins, such as Rho GTPase, or recruit new effectors, including IP3K and MAPK/ERK kinases, though β-arrestins with a G-protein-independent mechanism [75].

Muscarinic receptor activation promotes neuronal and glial cell differentiation. Cholinergic receptors are expressed in PNS. Immunocytochemistry studies showed muscarinic receptor expression on chicken and rat DRGs neurons, satellite cells, and myelinating and non-myelinating SCs. Muscarinic receptors are more abundant in the first phase of gangliogenesis [76]. M2 and M4 receptors are also expressed in peripheral sensory fibres, which are responsible for nociceptive stimuli perception and their transmission to the spinal cord [76]. During embryo development, ACh induces neurite outgrowth in chicken embryo DRG sensory neurons and increases neurofilament and transcriptional factors involved in the neuronal differentiation, such as c-jun and c-fos [77].

Muscarinic receptor activation also modulates the proliferation and survival of glial cells, such as astrocytes [78], oligodendrocytes [38], and SCs [40,41,42,68]. 

Oligodendrocytes (OLs) express muscarinic receptors [37,79,80] whose activation triggers different signal transduction pathways [80]. Gene expression analysis and immunostaining showed that the expression of muscarinic receptors changes during the maturation of oligodendrocytes, suggesting different subtype roles during their differentiation [37]. M1, M3, and M4 receptors are the main subtypes of AChRs expressed in OPCs (oligodendrocyte progenitor cells), whereas all muscarinic receptor subtypes are found at lower levels in mature OLs, implying that muscarinic receptors may be mainly required in the early stages of OL development [38]. Exposure to muscarine enhanced OPC proliferation, with an effect mainly mediated by M1, M3, and M4 receptors, as demonstrated by pharmacological competition binding experiments. This result is further supported by the decreased expression of MBP after muscarine exposure, which impairs their terminal differentiation towards myelinating phenotype. In support of these observations, another research group demonstrated that the muscarinic receptor antagonist, benztropine, promotes oligodendrocyte differentiation, favoring the rescue of the lesions in white matter in EAE mice [39].

### 4.1. Functional Expression of the M2 Acetylcholine Muscarinic Receptor in SC Proliferation

SCs’ main functions contribute to the axon–glial network during development [4,81]; to the electrical isolation of axons, allowing saltatory conduction [82]; and to post-injury nerve repair [83,84]. SCs are generated through two well-described intermediate stages: SC precursors (SCPs) in early embryonic nerves (mouse E12–13; rat E14–15) and immature SCs (iSCh, mouse nerves E15–16; rat nerves E17–18 extend to the perinatal period) in late embryonic and perinatal nerves [85]. During terminal differentiation, SCs can differentiate in myelinating SCs, which enwrap axons with a lipid-rich myelin sheath, promoting saltatory conduction, or in Remak SCs, which ensheath smaller unmyelinated axons [86] and perisynaptic SCs (PSCs) at the neuromuscular junction (NMJ) [87]. After a nerve injury, myelinating SCs and Remak SCs are rapidly activated by injury-induced, c-jun-mediated signals, undergoing dynamic cell reprogramming aimed at promoting the repair phenotype [88]. Moreover, PSCs rapidly respond to nerve injury and PSC mAChRs downregulation seems to be a general approach to facilitate NMJ repair in adulthood [89].

In the PNS, Villegas was the first who investigated this signalling on the nerve fibres of invertebrates, demonstrating that ACh receptors on SCs localise to the axon–SC interface [90,91]. Numerous studies describe ACh as being involved in the myelination process, but its specific role in myelinating glial cells remains poorly understood as a consequence of the huge complexity of the cholinergic system, the variety of ACh receptors types, and of the enzymes regulating ACh homeostasis. 

Cultures of SCs, obtained from sciatic nerves of 2-day-old neonatal rats, express M1, M2, M3, and very low levels of M4 muscarinic receptor subtypes. M2 mAChRs is the most abundant subtype and persists in mature SCs [40].

SCs treated with ACh mimetics have shown significant changes in intracellular concentrations of IP3 and cAMP levels [40]; the selective activation of M2 muscarinic receptors, by using the agonist arecaidine propargyl ester (APE), inhibits cell proliferation, causing a cell accumulation in the G1 phase of the cell cycle [41]. The arrest in the G1-S phase is followed by the modulated expression of several cell-cycle markers, with a decrease in PCNA levels and an increase in p27 and p53 proteins [41]. Moreover, the downregulation of c-jun, Notch-1, and Jagged-1 was also observed [42]. 

The co-treatment of APE and gallamine, a M2 preferential antagonist, rescues cell proliferation, supporting the evidence that cell growth decrease is M2 receptor-mediated. The cell cycle arrest is reversible because the removal of the M2 agonist from the culture medium is sufficient to reactivate cell division [41], thus preserving the well-known SC plasticity.

### 4.2. M2 Muscarinic Receptors Influence SC Myelinating Phenotype

Immature SCs take part in the radial sorting process that generates SCs at a 1:1 ratio with a segment of the larger diameter axons [81]. In rodents, these cells are considered as pro-myelinating cells, and they subsequently form myelin sheaths in the peri- and post-natal period. The smaller and unmyelinated axons are gradually associated with immature SCs forming Remak bundles. Individual cell fate is strictly determined by axonal–glia interaction through very complex signalling pathways [92]. Myelin and Remak cells are mitotically quiescent in healthy adult nerves but they can briefly re-enter the cell cycle after nerve injury [93]. As mentioned before, ACh, acting through the M2 receptor, reduces cell growth in a reversible manner [41], preserving SC proliferative capability. Although a reversible cell proliferation block, namely the activation of M2 receptors by the selective agonist APE, promotes SC myelinating differentiation with an early upregulation of the promyelinating markers Sox10 and Egr2 (Krox20), both at transcript and protein levels; this event is abolished by the use of the M2 antagonist, gallamine [42]. 

The M2 muscarinic receptor also controls Notch-1 signalling, which is involved in the transition from precursor to immature SCs [94]. Although the full length of Notch-1 is not modulated by M2 agonist treatment, the active form of Notch (Notch Intracellular Domain; NICD) is significantly decreased, followed by a significant decrease in Hes-1 and Jagged-1 levels. The downregulation of Jagged-1 is directly controlled by Notch-1 downregulation and indirectly by M2 activation. Interestingly, NICD overexpression is able to counteract the decrease in Jagged-1 protein levels through M2 receptor mediation [42].

The M2 muscarinic receptor, by upregulating the promyelinating transcription factor Sox10/Egr2(Krox20) expression and antagonising c-jun and Notch-1 activities, provides a molecular mechanism which promotes the exit of SCs from the cell cycle and supports their differentiation [41,42].

Recently, the signalling pathway downstream of the M2 muscarinic receptor responsible for the described effects was elucidated. M2 receptor activation negatively modulated the PI3K/Akt/mTORC1 pathway whose levels are relevant in the controlling of SC proliferation [75]. The ability of the M2 receptor to downregulate the levels and activity of the mTORC1 complex was confirmed by the decreased expression of its specific target, namely p-p70-S6K-Thr389. Moreover, the M2-mediated reduced expression of p-AMPKα-thr172, a negative regulator of myelination, confirms the role of the M2 receptor as a co-modulator of SC differentiation [75] (Figure 1). 

### 4.3. M2 Receptors Modulate SC Morphology

Morphological analysis shows that SCs in growth conditions have a classic bipolar morphology, with some rounded cells, typically associated with cells in the mitotic phase. According to Loreti and co-author [41], M2 activation causes a lack of mitotic cell morphology and SCs appeared juxtaposed to one another [42] with N-CAM and N-cadherin clustered in contact areas between neighbouring cells and the organisation of actin-stress fibres. Cytoskeleton and adhesion molecules redistribute in order to stabilise the new differentiative phenotype and myelin structures. Together with the morphological changes, the M2 receptor upregulates the expression of myelin proteins MBP, P0, and PMP22, both at transcript and protein levels [42]. Considering that the non-selective muscarinic agonist muscarine, activating all AChR on SCs, is able to faintly upregulate only PMP22, without any effect on the expression of the other peripheral myelin proteins (MBP, P0), it is possible to conclude that the enhancement of the SC differentiation towards myelinating phenotype is exclusively M2 receptor-mediated [42]. 

The ultrastructural analyses of M2/M4^−/−^ mice by electron microscopy (TEM) demonstrated that M2/M4^−/−^ sciatic nerves have more degenerating axons and alterations in myelin organisation of the medium/large axons when compared with wild type (WT); moreover, the presence of the myelin sheath with different degrees of compactness and irregular ensheathments and progressive myelin sheath disorganisation were also observed [42]. Since it has been demonstrated that the M4 receptor is almost absent in SCs [40], the observed myelin alterations can be ascribed to the lack of M2 receptors.

More recent data have also demonstrated that the effects mediated by M2 muscarinic receptors in the rodents can be translated in human SCs [68], where the M2 receptor is also present. The activation of M2 receptors with the preferential agonist decreases cell proliferation without altering survival. This is followed by a consequent upregulation of Egr2/Krox20 and addresses morphological changes with a more elongated morphology in APE-treated cells. Similar results observed between rat and human SCs support the idea that M2 receptor mechanisms are conserved across different mammals [95].

### 4.4. Cholinergic Signals, Mediated by Muscarinic Receptors, Modulate Nerve Growth Factor Production and Release

PNS regeneration after injury is almost entirely mediated by SCs, given their ability to produce several growth factors (e.g., brain-derived growth factor (BDNF), glial-derived growth factor (GDNF), and nerve growth factor (NGF)) [83,93,96]. In the CNS, a relationship also exists between ACh and neurotrophic factor expression. For instance, the cholinergic system is involved in the modulation of NGF and BDNF mRNA expression in the hippocampus [97]. After sciatic nerve damage, NGF and their receptors are upregulated in SCs, stimulating peripheral nerve response to damage; once reinnervation takes place, SCs myelinate the regenerating axons and neurotrophic factor levels and their receptors return to physiological levels [98]. 

The pharmacological modulation of muscarinic receptors regulates the production and secretion of NGF in SCs [43]. ELISA assay showed that the stimulation of muscarinic receptors significantly regulates NGF production, maturation, and release. Already after 24 h of M2 agonist or muscarine exposure, proNGF concentration was significantly increased in cell lysates, whereas the intracellular maturation of NGF was unchanged upon 48 h of cholinergic mimetics exposure [43]. Although a significant increase of proNGF release was observed upon 24 h of M2 agonist treatment, the level of mNGF concentration was significantly reduced only after 48 h. These findings are supported by a significant increase in tPA activity, involved in many processes, including the conversion of proNGF into mNGF [99]. Given that tPA activity is elevated during axonal outgrowth in the PNS [100,101], it follows that increased activity in SCs after M2 selective or non-selective agonists, may promote regenerating neurites in the injured environment. Moreover, it has been demonstrated that the M2 receptor significantly regulates the two recently discovered proNGF isoforms A and B, involved in survival and apoptotic processes, respectively. Interestingly, both selective M2 agonist and non-selective agonist muscarine treatments did not modify the expression of proNGF-A, whereas they significantly reduced proNGF-B expression, suggesting that the negative modulation of proNGF-B improves SC capability in counteracting neuronal cell death [43].

## 5. Nicotinic Receptors Involvement in Peripheral Nerve Injury and Inflammation

Nicotinic acetylcholine receptors (nAChRs) are members of the ligand-gated ion channels family that is composed by pentamers of hetero- or homo-subunits, forming a central cation-selective pore that is permeable to sodium, potassium, and calcium. Every subunit is formed by an N-terminal and a short C-terminal extracellular domain, four transmembrane domains (TM1-TM4), and a variable cytoplasmic loop. Two cysteine residues are present in the α subunit, which is important for ACh binding. The other subunits are indicated as non-α (β, γ, δ, ε). They are widely and differentially expressed in several regions of the CNS, PNS, and skeletal muscle [44]. 

During embryogenesis, the temporal coincidence of both ChAT and nAChR subunits demonstrated that ACh, via nAChRs, regulates different aspects of nervous system development [16]. 

nAChRs are also expressed in glial cells [16] and immune cells [44] where they have gained attention for the potential therapeutic targeting of inflammation by affecting the release of pro-inflammatory molecules in peripheral immune system cells as well as in microglial cells [102,103,104,105].

A prevalent expression of α7 nAChR is observed in astrocytes and microglia where it is involved in neuroprotection and is altered in neurodegenerative diseases, such as Alzheimer’ and Parkinson’s [44,106,107].

This receptor subtype is today considered a potential therapeutic target for several neurological diseases due to its activities in neuroprotection, synaptic plasticity, and neuronal survival. 

In PNS, after nerve transection, the distal segment undergoes a gradual process of degeneration, known as Wallerian degeneration. Both myelin and Remak cells have the ability to convert to a transient c-jun-mediated repair-promoting phenotype (*Repair SCs*) in response to the different environmental signals generated by nerve injury [83,93,108,109]. 

Axo–glial communication is essential for nerve repair; *Repair SCs* create a permissive environment for axon regrowth, but axon-derived signals also promote the later differentiation of *Repair SCs*. These cells stimulate a sequence of functions that allow myelin clearance, the downregulation of myelinating transcriptional factors, such as Egr2/Krox20 and myelin proteins (e.g., P0, MBP, MAG, and periaxin), counteracting neuronal death and upregulating the synthesis of several neurotrophic factors (i.e., NGF, BDNF, NT-3, NT-4/5, and NT-6) [83,110,111]. Moreover, in order to promote myelin and axon debris elimination, *Repair SCs* release pro-inflammatory cytokines that attract macrophages at the site of lesion. 

Considering the role of ACh in controlling SC differentiation, its absence or reduced levels, as observed in the in vitro proliferation recovery [41], help rescue cell proliferation. In addition, ACh ameliorates inflammation response via nAChR stimulation [112]. 

α7 nAChR is one of the most abundant subtypes expressed in the CNS, but it is also found in the PNS as well as in the immune system and several peripheral tissues [69]. After nerve injury, ACh has a relevant role in motor nerve terminal outgrowth and muscle repair, working on nAChRs. In fact, the block of nAChRs in the muscles by α-bungarotoxin (α-Btx) significantly inhibits axon outgrowth [113].

This suggests that local release of ACh is essential and the activation of muscarinic or nicotinic receptors directly triggers nerve terminal outgrowth and/or modulates the release of neurotrophic factors. Moreover, nAChRs modulate the turnover of ACh given their ability to work as ACh sensors outside the synaptic space, thus regulating ACh degradation [45].

Injured sciatic nerves show a significant upregulation of α7 nAChRs up to 5 days after nerve injury, with the peak at 3 days. An intense immunoreactivity for α7 nAChRs in SCs in a rat nerve crush injury model has also been demonstrated. TNF-α is elevated in the first 24 h, then decreases and returns to a normal level within 5 days. It appears that the inflammatory response is suppressed by the upregulation of nAChRs, which has been demonstrated as an important receptor to prevent cytokine release after its activation [114]. 

Moreover, the application of selective agonist PNU 28298 decreases TNF-α levels and promotes axonal regeneration; the administration of PNU 28298 significantly enhances regeneration distance, with a greater number of unmyelinated and myelinated regenerated fibers per unit area. These events are counteracted by the use of a selective α7 antagonist, methyllycaconitine, suggesting a key role of α7 nAChRs in the inhibition of peripheral inflammation, the promotion of regeneration, and rescue of the tissue homeostasis [46].

More recently, using an in vitro model of sciatic nerve degeneration, it has been demonstrated that the activation of α7 nAChRs by a new selective agonist, ICH3, was able to negatively modulate IL-6 synthesis and release in SCs, contributing to a significant decrease in peripheral inflammation. Moreover, it is also able to modulate the activity of metalloproteinases 2 and 9 (MMP2 and MMP9, respectively) [47]. This latter effect is relevant considering that MMP2 and MMP9 activities are required for extracellular matrix (ECM) remodelling and for the extracellular maturation and degradation of NGF [43,115].

Although further analyses are necessary to fully clarify the role of ACh and α7 nAChRs in PNS and in neuroinflammation, in vitro study suggests that SCs are able to synthesise low levels of ACh when cultured in absence of axons, confirming its possible alternative role, mediated by α7 nAChRs, during nerve regeneration [116]. Moreover, the dysregulation or disruption of the nAChRs-mediated anti-inflammatory ACh pathway is observed in demyelinating disorders, such as multiple sclerosis (MS) and other neurodegenerative diseases. For example, in MS, the progression of the pathology is associated with the loss of myelin caused by decreased oligodendrocyte differentiation and their incapacity to remyelinate. Experimental autoimmune encephalomyelitis (EAE), chronically treated with an AChE inhibitor, presents a reduction of neuroinflammation and demyelination accompanied by the improvement of the disease symptoms; effects that are abolished by α7 nAChR antagonists [117,118]. Moreover, the treatment with nicotine significantly reduced the inflammatory and autoimmune response in EAE. In the periphery, nAChRs inhibit the proliferation of the autoreactive T-cells, altering the cytokine profile. 

In accordance with these results, the reduced levels of ACh accompanied by the increased activity of cholinesterase enzymes (i.e., AChE and BuChE) were measured by enzymatic analysis both in the serum and cerebrospinal fluid of MS patients, demonstrating that upregulated levels of these enzymes may be responsible for reduced ACh levels [119]. It seems evident that the cholinergic anti-inflammatory pathway by α7 nAChRs activation may play a strategic role in this neuropathology.

## 6. Conclusions

The last thirty years has been a period of advancement in the identification of neurotransmitter receptors’ involvement in glial cell physiology, demonstrating their alternative role in the cross-interactions between glial cells and neurons. In this review, we presented some of the most recent findings on neurotransmitter signalling in glial development and differentiation. In the last decades, the literature has shown that many different receptors can regulate the same glial physiological events. However, this redundant and complex integrated regulation system could be explained as a mechanism of preservation of glial cell physiology; in case of a single receptor signalling dysfunction, other neurotransmitters can overcome the deficit, preserving the functions of glia and the health of the nervous system. 

We focused our attention in particular on ACh and its receptors in SC development and function, highlighting the ability of this neurotransmitter to preserve SC plasticity and at the same time to address SC differentiation. We emphasised the consolidated role of M2 muscarinic receptors in the control of proliferation and in the promotion of cell differentiation towards the promyelinating phenotype (Figure 2); moreover, treatments with cholinergic mimetics positively modulate NGF, underlining the possible role of the M2 receptor in the preservation of neuronal and/or glial components after axonal damage. The ability of M2 receptors to modulate neurotrophic factor production and release suggests their potential role in promoting the regenerative capability of SCs in the PNS [43,120].

At the same time, α7 nAChR expression is also relevant. In particular, its increased expression in SCs after peripheral nerve injury highlights its potential role in the modulation of both the inflammatory microenvironment and axon regeneration [47] (Figure 2).

Cholinergic receptors expressed in SCs can be activated by ACh released in the extra-synaptic regions, along the sensory and motor axons [19]; however, it has also been demonstrated that SCs can produce low levels of ACh when cultured in the absence of axons, suggesting that low ACh levels may contribute to maintain cholinergic activity in SCs after peripheral axonal injury [116]. 

Altogether these findings describe a central role of ACh and cholinergic receptors in SC differentiation and plasticity, suggesting that they could represent a putative target for identifying novel pharmacological treatments in order to speed up the regeneration process and to inhibit central and peripheral neuroinflammation [47,119]. 

Cholinergic receptors are impaired in a number of nervous system disorders (e.g., schizophrenia, Alzheimer’s disease, and neurodevelopmental pathologies) and for these reasons, the identification of new drugs regulating these functions might be of great clinical relevance. Although some cholinergic drugs are already used as therapeutic approaches in several pathologies (e.g., benztropine in Parkinson’s disease or xanomeline and cevimeline in Alzheimer’s disease), the large distribution of these receptors in several tissues and the limited availability of selective ligands have restricted therapies using cholinergic receptors as targets [121]. Today, the increased knowledge in medicinal chemistry and in bioinformatics accompanied by drug delivery studies might open a fascinating therapeutic perspective for cholinergic mimetics for the treatment of several nervous system pathologies and in reducing the neuroinflammation both in the central and peripheral nervous systems [122]. 

## Figures and Tables

**Figure 1 biomedicines-11-00041-f001:**
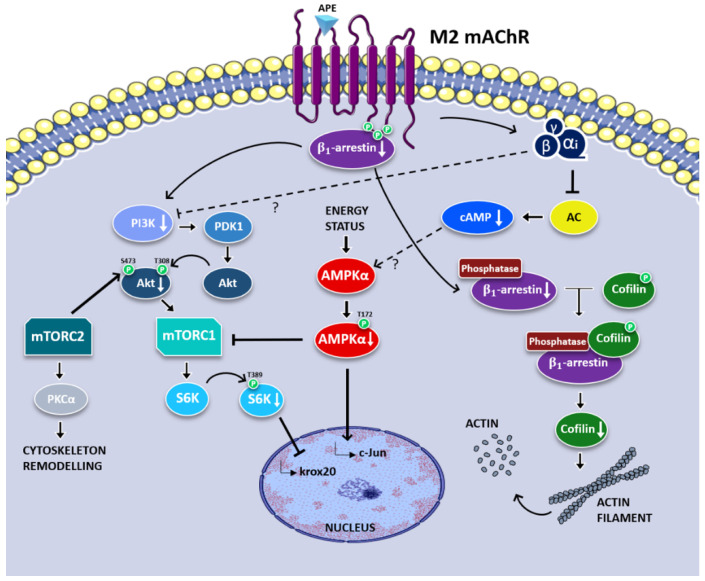
The signal transduction pathways activated downstream of the M2 muscarinic receptors mainly involve G_αi_ with cAMP level reduction. However, M2 receptors can also negatively modulate the PI3K/AKT/mTORC1 pathway via a β1-arrestin decrease or βγ subunits. All these signals may contribute to negatively modulate SC proliferation and to promote their differentiation [75].

**Figure 2 biomedicines-11-00041-f002:**
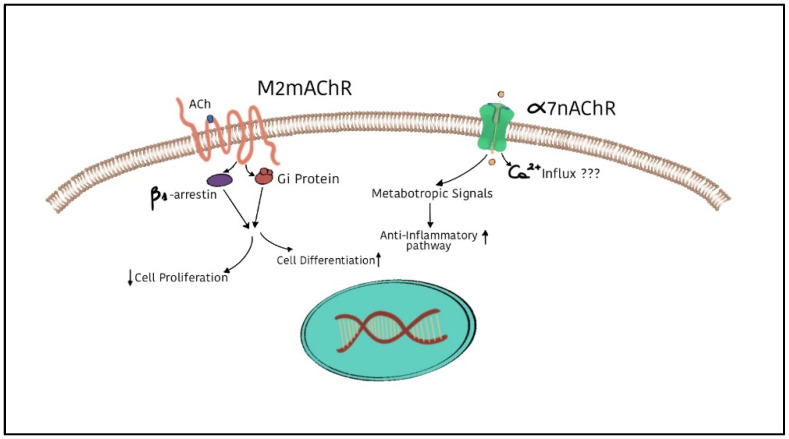
Schematic representation of the effects mediated by M2 muscarinic receptors and α7 nicotinic receptors on Schwann cells during SC development or after nerve damage, respectively. Metabotropic signals downstream of M2 receptor activation inhibit SC proliferation and promote their differentiation. However, α7 nicotinic receptors, expressed by SCs after peripheral fiber injury, contribute to modulating the anti-inflammatory pathway, thus restoring cell homeostasis.

**Table 1 biomedicines-11-00041-t001:** Neurotransmitter expression and function in glial cells.

*Neurotransmitters*	*Receptors*	*Functions*	*References*
*Adenosine/ATP*	Both types	Oligodendrocytes development and myelin conservation.	[15]
	P1	Oligodendrocytes differentiation and promotion of myelination.	[20]
	P2X7	In KO mice, reduced expression of P0 and MAG; increased in Remak bundles. Increased expression in microglia and astrocytes after trauma.	[21,22,23]
	P2X4	Increased expression in microglia after trauma. In SCs, strongly upregulated after injury, enhances motor and sensory recovery and myelination.	[24,25]
*Glutamate*		Glutamate regulates proliferation, migration, and differentiation of OPCs and remyelination after damage.	[26,27]
	AMPA/KAR	c-fos, c-jun, and jun-B modulation.	[28]
	AMPA/KAR	OPC proliferation and differentiation blocking.	[16,29,30,31]
	mGluR2	Negative regulation on myelination.	[32]
*GABA*		Related to myelin biology in CNS.	[33]
	GABA-A	Controls the levels of GAD67.	[34]
	GABA-B	Highly expressed in non-myelinating SCs with downregulation of P0 and PMP22 expression. KO mice showed effect on axonal size.	[35,36]
*Acetylcholine*	Muscarinic receptors	Their expression changes during OPC maturation.	[37]
	M1, M3, M4 muscarinic receptors	Enhance OPC proliferation and decrease MBP levels. Antagonism by benztropine promotes oligodendrocyte differentiation in EAE mice.	[38,39]
	M2 muscarinic receptor	Most abundant muscarinic receptor subtype in SCs. Reversible arrest of the cell cycle and cell accumulation in the G1 phase with increased expression of Egr2 and myelin protein. M2/M4^−/−^ mice show myelin alteration. Promotes the production and secretion of NGF with downregulation of proNGF-B.	[40,41,42,43]
	Nicotinic receptors	Potential targets in inflammation in glial cells and immune cells. They are involved in cognitive functions and are compromised in Alzheimer disease.	[16,44]
	α7 nAChR	Works as an ACh sensor in the synaptic space, regulating ACh degradation. Inhibits peripheral inflammation and rescues tissue homeostasis. Modulator of neuroinflammation.	[45,46,47]

ACh: acetylcholine; CNS: central nervous system; EAE: experimental autoimmune encephalomyelitis mouse model; Egr2: early growth response 2; GAD67: glutamic acid decarboxylase; MAG: myelin-associated glycoprotein; MBP: myelin basic protein; NGF: nerve growth factor; OPC: oligodendrocyte progenitor cells; P0: myelin protein 0; PMP22: peripheral myelin protein 22; Pro-NGF: nerve growth factor precursor; SCs: Schwann cells.

## Data Availability

Not applicable.

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
