# Peer review of "Emerging Roles of Cholinergic Receptors in Schwann Cell Development and Plasticity"

_biomedicines, 2022, doi:10.3390/biomedicines11010041_

Round 1
Reviewer 1 Report
The review is comprehensive and relevant. The review article includes a clear and concise abstract. The introduction sets the scene by describing all the recent finding to uncover the perspective of alternative functions of neurotransmitters in glial cells in the context of cholinergic receptors in Schwann cell development and plasticity. The key message is through text, figure 1 and conclusions. However, some more information about the activation of M2 receptors with a selective agonist in regard to upregulation of Egr2/Krox 20 and morphological changes in human Schwann cell should be added. Also please provide some more details on dysregulation or disruption of ACh anti-inflammatory pathway in Multiple Sclerosis. The methodology for literature search is not described. The spell check and grammar should be performed.
Author Response
A better explanation on the role of M2 receptors in human Schwann cell differentiation has been added to page 7. More details on ACh dysregulation in multiple sclerosis have also added to page 10.
Reviewer 2 Report
This review comprehensively summarized the data reporting the functional role of neurotransmitters in regulating glial cell development and functions. This is a timely review discussing the emerging role of cholinergic receptors in Schwann cell development and plasticity. However, I have some concerns that need to be addressed:
1. It would be better if the authors provided the search strategies for this review.
2. In addition to presenting the data of previous work, the authors should underline the new findings and meanings which may make this review remarkable and interesting to the audience of the present journal. Therefore, it is recommended to extend the discussion and conclusion sections. For instance, the possible therapeutic strategy targeting neurotransmitter receptors in the treatment of relevant diseases should be described and discussed deeply, including current ongoing preclinical studies or trials. Also, a discussion of possible future research for choline receptors could be enhanced.
3. The improve the manuscript, the authors should sharpen the text and try to avoid overly verbose expressions. e.g., "among the huge variety of molecules mediating these interactions...the development and physiology of the glial cells" in the abstract part. In its current version, the manuscript is hard to read.
4. The authors mentioned, "Schwann cell", while sometimes using "SC". Try to use the full terms when they first appear in the manuscript.
5. In Table 1, the author summarized “P2X7-Increased expression in microglia and astrocytes after trauma” “P2X4-Increased expression in microglia after trauma” in Functions Part. However, these descriptions are not the receptors’ functions, or the authors could replace “Functions” with a more appropriate expression. In addition, changing “Both types” to “P1/P2” may be better. I recommend the author list the abbreviations of the words involved in Table 1 at the bottom of the table for a better understanding.
6. Is there anything missing in Section 5 "Nicotinic receptors", only 5.1 "α7 ACh nicotinic receptors in peripheral nerve injury and inflammation" can be seen in this section.
Author Response
This review comprehensively summarized the data reporting the functional role of neurotransmitters in regulating glial cell development and functions. This is a timely review discussing the emerging role of cholinergic receptors in Schwann cell development and plasticity. However, I have some concerns that need to be addressed:
1. It would be better if the authors provided the search strategies for this review.
2. In addition to presenting the data of previous work, the authors should underline the new findings and meanings which may make this review remarkable and interesting to the audience of the present journal. Therefore, it is recommended to extend the discussion and conclusion sections. For instance, the possible therapeutic strategy targeting neurotransmitter receptors in the treatment of relevant diseases should be described and discussed deeply, including current ongoing preclinical studies or trials. Also, a discussion of possible future research for choline receptors could be enhanced.
Thank you for your comments. The manuscript has been changed in several parts and the conclusions were expanded. Detailed informations on preclinical studies in particular for muscarinic receptors are reported in previous review (see Tata, 2008, Matera and Tata 2014). The main focus of this review was to highlight the new roles of ACh in Schwann cell development and function. At least there are not preclinical studies for the treatment of Schwann cells dysfunction with cholinergic ligands.
3. The improve the manuscript, the authors should sharpen the text and try to avoid overly verbose expressions. e.g., "among the huge variety of molecules mediating these interactions...the development and physiology of the glial cells" in the abstract part. In its current version, the manuscript is hard to read.
We have changed several parts in the abstract
4. The authors mentioned, "Schwann cell", while sometimes using "SC". Try to use the full terms when they first appear in the manuscript.
We have corrected the text and used abbreviations after the first citation of the full term.
5. In Table 1, the author summarized “P2X7-Increased expression in microglia and astrocytes after trauma” “P2X4-Increased expression in microglia after trauma” in Functions Part. However, these descriptions are not the receptors’ functions, or the authors could replace “Functions” with a more appropriate expression. In addition, changing “Both types” to “P1/P2” may be better. I recommend the author list the abbreviations of the words involved in Table 1 at the bottom of the table for a better understanding.
We have revised the table 1.
6. Is there anything missing in Section 5 "Nicotinic receptors", only 5.1 "α7 ACh nicotinic receptors in peripheral nerve injury and inflammation" can be seen in this section.
We have eliminated the paragraph 5.1 and collected the two paragraphs in only one.
Reviewer 3 Report
Review article titled (Alternative functions of Neurotransmitters in glial cells: emerging role of Cholinergic Receptors in Schwann Cell Development and Plasticity) by Piovesana et al. discusses the potential importance of glia cells Schwann cells development as well as plasticity. They focused on the role of cholinergic receptors. I find this review is useful but some revisions are necessary.
1- Title : needs revision to start with cholinergic receptos as this is the main focus of the review.
2- I suggest the authors may design a graphic abstract or diagram to express the topic of the review
3- It is necessary to provide evidence from animal experiments or in vitro studies on the importance of both the cholinergic M & N receptors in disease models & how modulation by drugs could modulate the disease state.
Author Response
Review article titled (Alternative functions of Neurotransmitters in glial cells: emerging role of Cholinergic Receptors in Schwann Cell Development and Plasticity) by Piovesana et al. discusses the potential importance of glia cells Schwann cells development as well as plasticity. They focused on the role of cholinergic receptors. I find this review is useful but some revisions are necessary.
- Title : needs revision to start with cholinergic receptos as this is the main focus of the review.
The title has been modified
- I suggest the authors may design a graphic abstract or diagram to express the topic of the review
We have added an other figure that we have also used for graphical abstract
- It is necessary to provide evidence from animal experiments or in vitro studies on the importance of both the cholinergic M & N receptors in disease models & how modulation by drugs could modulate the disease state.
Information about the effects mediated by cholinergic ligands (both antagonists and agonists) for muscarinic and nicotinic receptors were tested in EAE model to analyse their effects on the oligodendrocyte differentiation and for the modulation of neuroinflammation, respectively. These results have been reported in paragraph 4 and 5 of the revised text. The large part of the results described have been obtained using in vitro models.
Round 2
Reviewer 3 Report
The revised version of the review "Emerging roles of cholinergic receptors in Schwann Cell development and plasticity" by Piovesana et al. was improved and acceptable in the current form.